

# Impact of COVID-19 outbreak on the mental health status of undergraduate medical students in a COVID-19 treating medical college: a prospective longitudinal study

Ilango Saraswathi[1,*], Jayakumar Saikarthik[2], K. Senthil Kumar[1,*], Kumar Madhan Srinivasan[3], M. Ardhanaari[4] and Raghunath Gunapriya[5]

[1] Department of Physiology, Madha Medical College and Research Institute, Chennai, Tamil Nadu, India
[2] Department of Basic Medical Sciences, Majmaah University, Ar-Riyadh, Saudi Arabia
[3] Department of General Medicine, Madha Medical College and Research Institute, Chennai, Tamil Nadu, India
[4] Department of Psychiatry, Meenakshi Medical College Hospital and Research Institute, Enathur, Tamil Nadu, India
[5] Department of Anatomy, Saveetha Medical College Hospital, Thandalam, Tamil Nadu, India
* These authors contributed equally to this work.

Corresponding author
Jayakumar Saikarthik,
s.jaya@mu.edu.sa

## ABSTRACT

**Background:** The COVID-19 pandemic is found to affect the mental health of the population. Undergraduate medical students are especially prone to mental health disorders and hence could be more vulnerable to the impact of the pandemic.
**Methods:** A prospective longitudinal study was conducted on 217 undergraduate medical students in a medical college at Chennai, India. Depression, anxiety, and stress levels were recorded using Depression Anxiety Stress Scale 21 Items (DASS21) before and during the COVID-19 outbreak in India in December 2019 and June 2020, respectively. In the follow-up survey, in addition to DASS21, the Pittsburgh Sleep Quality Index to assess sleep quality and a self-administered questionnaire to assess the impact of COVID-19 related stressors were used. The self-administered questionnaire assessed the status of COVID-19 testing, interactions with COVID-19 patients, self-perceived levels of concerns and worries related to academics (COVID-19-AA (academic apprehensions)) and those pertaining to the self and family/friends (COVID-19-GA (general apprehensions)). Cross-sectional and longitudinal comparison of overall scores of depression, anxiety, and stress and scores stratified by gender, year of study, place of residence and monthly family income were performed. Predictors for depression, anxiety, and stress during COVID-19 were investigated using adjusted binary logistic regression analysis and results were expressed as adjusted odds ratio with 95% confidence interval (CI). A $P$ value < 0.05 was considered statistically significant.
**Results:** The average scores of depression, anxiety, and stress during the baseline survey were 7.55 ± 7.86, 4.6 ± 6.19 and 7.31 ± 7.34 with the prevalence (95% Cl) of 33.2% [27–39.9%], 21.2% [16–27.2%] and 20.7% [15.5–26.7%]; in follow-up survey, the mean scores were 8.16 ± 8.9, 6.11 ± 7.13 and 9.31 ± 8.18 with the
prevalence being 35.5% [29.1–42.2%], 33.2% [27–39.9%] and 24.9% [19.3–31.2%] for depression, anxiety, and stress respectively. There was a significant increase in both the prevalence and levels of anxiety and stress ($P < 0.001$), with depression remaining unchanged during COVID-19, irrespective of gender, year of study, place of residence and family's monthly income. Poor sleep quality, higher levels of baseline depression, anxiety, and stress, higher COVID-19-GA, COVID-19 patients in family/friends and direct interactions with COVID-19 patients were found to be significant predictors of negative mental health in undergraduate medical students. COVID-19-AA was not significantly associated with depression, anxiety, and stress. **Conclusion:** The COVID-19 pandemic appears to negatively affect the mental health of the undergraduate medical students with the prevalence and levels of anxiety and stress being increased, and depression symptoms remaining unaltered. Addressing and mitigating the negative effect of COVID-19 on the mental health of this population is crucial.

# INTRODUCTION

The World Health Organization (WHO) announced COVID-19 outbreak initially, as a public health emergency of international concern (PHEIC) on January 30, 2020 and later declared as a pandemic on March 11, 2020 (*WHO, 2020*). In India, the first COVID-19 case was reported in Kerala on January 30, 2020, and by May 19, the number of cases had crossed one hundred thousand. By September 7, India became the world's second worst hit nation with 4.2 million confirmed COVID-19 cases following United States of America and has recorded 71,642 deaths (*The New York Times, 2020*). Within India, the state of Maharashtra was the worst hit state followed by Andhra Pradesh and Tamil Nadu, contributing to 21.6%, 11.8% and 11% of the total cases respectively (*Newsdesk, 2020*). The Government of India declared a nationwide lockdown on 25th March 2020, as a measure to mitigate the spread of infection. However, prolonged lockdown is not only unfavorable to the individuals, it also significantly affects the nation's economy. As a way to revive and restore the affected economy, a phase-wise upliftment of lockdown was announced from June 1, easing some restrictions, while the lockdown was maintained for the containment zones alone.

Public health emergencies during epidemic/pandemic like SARS, MERS and Ebola outbreak were associated with increased psychological distress in the affected population (*Batawi et al., 2019*; *Lee et al., 2007*; *Lotsch et al., 2017*). Maladaptive behaviors, emotional and defensive reactions were some of the psychological responses to pandemic (*Taylor, 2019*). Social isolation was found to be strongly associated with anxiety, depression, self-harm, and suicidal tendencies (*Matthews et al., 2019*). Studies indicated that social distancing for a longer duration could affect the mental health negatively (*Reynolds et al., 2008*). Isolation, boredom, frustrations, worries about contracting the

infection, lack of freedom, concerns for family/friends are some of the factors that could affect mental well-being during quarantine (*Brooks et al., 2020*). Poor sleep quality and increased psychological distress were also well-documented during earlier pandemics (*Chen et al., 2006*; *Johal, 2009*). In particular, poor sleep was associated with negative emotions, depressive symptoms and increased risk of mental illness (*Agargun, Kara & Solmaz, 1997*; *Tao et al., 2017*).

In a recent study conducted during the COVID-19 outbreak in India, one fifth of adults were found to suffer from depression and stress and one fourth from anxiety (*Saikarthik, Saraswathi & Siva, 2020*). The mental health of medical students was found to be even poorer, when compared to general population (*Bergmann, Muth & Loerbroks, 2019*). Medical education is the most demanding of all the other professional programs in terms of both academics and emotional component of the students (*Wolf, 1994*). Globally, one in three medical students were found to have anxiety, which was higher than the general population (*Tian-Ci Quek et al., 2019*). The level of depression, suicidal ideation, suicide rates, substance abuse and mental health disorders were also found to be higher among medical students (*Hays, Cheever & Patel, 1996*; *Molodynski et al., 2020*; *Schwenk, Davis & Wimsatt, 2010*). Although medical students have better access to mental health care, they were less likely to seek mental health help compared to general population, mainly due to stigma surrounding mental health disorders. This may lead to untoward and harmful coping methods like excess alcohol consumption and substance abuse (*Chew-Graham, Rogers & Yassin, 2003*; *Rosenthal & Okie, 2005*).

The swine flu (H1N1) outbreak in 2009 was the last outbreak of an infectious disease in a pandemic scale to which India was exposed (*WHO, 2009*). Undergraduate medical students in India are usually in the age group of late teens to mid-twenties, and hence the current COVID-19 outbreak is the first exposure to them as adults on a pandemic level. In addition, medical students are facing challenges such as sudden changes in their training routine, including teaching and assessment via online sessions, decreased patient contact and interactions with peers to name a few. These changes result in increased screen time, possible hinderance to their training and increased risk of contracting the infection mainly among the students in clinical postings. All these factors could eventually exert a toll on the mental and emotional well-being of the medical students as they are on an unknown territory.

Earlier studies show that the negative impact of epidemic/pandemic on the mental health are higher in healthcare workers (*Lee et al., 2018*; *Lu, Shu & Chang, 2006*; *McAlonan et al., 2007*). Unfortunately, only limited studies were done on the impact of epidemic/pandemic on the mental health of medical students. Studies on the impact of the COVID-19 pandemic on medical students are limited to cross-sectional surveys assessing attitude, awareness, knowledge, precautionary measures, concerns, risk perceptions, impact on education and confidence, and fear of COVID-19 (*Agarwal et al., 2020*; *Ahmed et al., 2020b*; *Choi et al., 2020*; *Khasawneh et al., 2020*; *Nguyen et al., 2020*; *Taghrir, Borazjani & Shiraly, 2020*). Literature search showed only a single study about the psychological impact of COVID-19 on medical students, which cross-sectionally assessed their anxiety levels (*Cao et al., 2020*).

Globally, few longitudinal studies compared mental health before and during COVID-19, and found an increase in anxiety and depression symptoms in college students in China (*Li et al., 2020a*), the United States (*Huckins et al., 2020*) and a deterioration of mental health in the general population in the United Kingdom (*Pierce et al., 2020*). To our knowledge, there are no studies analyzing the impact of the COVID-19 outbreak on the mental health of undergraduate medical students prospectively to assess cause and relationship. From these observations, we hypothesized that the COVID-19 outbreak and quarantine would have a serious negative impact on the mental health of undergraduate medical students. Hence, we conducted a prospective longitudinal study to investigate the mental health of undergraduate medical students over a duration of 6 months by analyzing data collected before and during the COVID-19 outbreak in India. The study was conducted in a medical college in Chennai, Tamil Nadu, which is a center for treating COVID-19 patients. We did an extensive investigation of possible confounders and predictors of mental health disorders including demographics, sleep quality, apprehensions related to and caused by COVID-19 in terms of academics and concerns for the self, family, friends, and interpersonal relationships.

# MATERIALS AND METHODS

## Participants and setting

The study was originally planned to be a cross-sectional survey to assess the mental health of the undergraduate medical students in the institution. There were 300 medical students in the institution enrolled for undergraduate medical degree. The Bachelor of Medicine and Bachelor of Surgery (M.B.B.S) is a 5.5 years undergraduate medical course offered in India in which the first 2.5 years concentrate mostly on basic medical sciences (pre and para-clinical subjects) and the next 2 years on clinical subjects followed by 1 year of Compulsory Rotatory Resident Internship (CRRI). All 300 students studying in pre-clinical (1st year), para-clinical (1.5 years after pre-clinical year) and clinical years (pre-final year, final year, and resident interns) were included for the study and convenience sampling method was used. The students were explained about the objective of the study and were informed that the participation was voluntary, and confidentiality will be maintained. A total of 276 students out of the total 300 students agreed to take part in the study from whom written consent was obtained before the beginning of the study.

## Baseline (Before COVID-19) survey

Basic sociodemographic details such as age, gender, year of study, area of current residence and gross monthly income of the family were collected, and the mental health status was assessed using Depression Anxiety Stress Scale 21 items (DASS21). Students below 18 years of age and those with self-reported history of any pre-existing chronic medical conditions including mental health disorders were excluded (five were underage and two reported history of bronchial asthma). The remaining 269 participants who were included for the study were contacted during their free time, after classes and were encouraged to answer the survey sincerely and doubts were clarified. Email id and mobile number were

collected from all the participants. This part of the study was conducted during the first 2 weeks of December 2019, which was before COVID-19 outbreak in India.

### Follow-up (During COVID-19) survey

With the unexpected changes to normalcy caused by the COVID-19 outbreak and subsequent lockdown, the authors decided to prospectively study the mental health status of the medical students to assess the effects of COVID-19 pandemic on mental health of the study population. After obtaining permission from Institutional Ethics Committee (IEC), the original data from the cross-sectional study was decided to be taken as "before COVID-19 data" (baseline) and another survey was conducted on June 2020 (June 10–20) to collect "during COVID-19" data (follow-up).

The follow-up survey was conducted via Google form whose link was sent through personal email IDs of the students which were collected during the baseline survey. This protocol was exercised in order to follow strict social distancing protocol and to avoid direct contact. The follow-up survey included five sections; first section had a detailed description of the purpose of the study, along with the informed consent. This section explained the importance and benefits of the survey in the current pandemic, highlighting the voluntary nature of participation and assurance of confidentiality of the collected data. Only after consenting to the study, the participants could access the remaining sections. The successive sections collected responses for demographic details, self-administered questionnaire, DASS21 and Pittsburgh Sleep Quality Index (PSQI). Out of the 269 participants from the baseline survey, 30 randomly selected students were included in a pilot study (described below) and were hence excluded from the follow-up survey. Out of the remaining 239 participants, 222 students responded, from which five responses were excluded because of being incomplete (response rate 90.8%). The final sample size of this prospective longitudinal study was 217. A flowchart illustrating the sample selection from baseline to follow-up survey is shown in Fig. 1.

### Survey instruments

To fulfill the objective of our longitudinal study, besides DASS21, "during COVID-19" data also included a self-administered questionnaire to assess the impact of COVID-19 related stressors and PSQI to assess the sleep quality of the students.

### Assessment of the impact of COVID-19 related stressors

A self-administered questionnaire was prepared by the authors after an extensive literature search, discussion with peers and local experts (*Wang et al., 2020*; *Wong, Gao & Tam, 2007*). It included 12 close-ended questions out of which, Items 1–3 focused on the subjects' status of COVID-19 testing (Yes/No) and their interactions with COVID-19 patients (Yes/No/I don't know). The remaining nine items were designed to assess self-perceived levels of concerns and worries for the self (4–6) and family/friends (7–8) and those related to academics (9–12), due to COVID-19 outbreak and quarantine (Supplemental File-Other). The responses were measured on a Likert scale of score 1 to 5, with 1 being the least and 5 being the maximum. This questionnaire was first tested

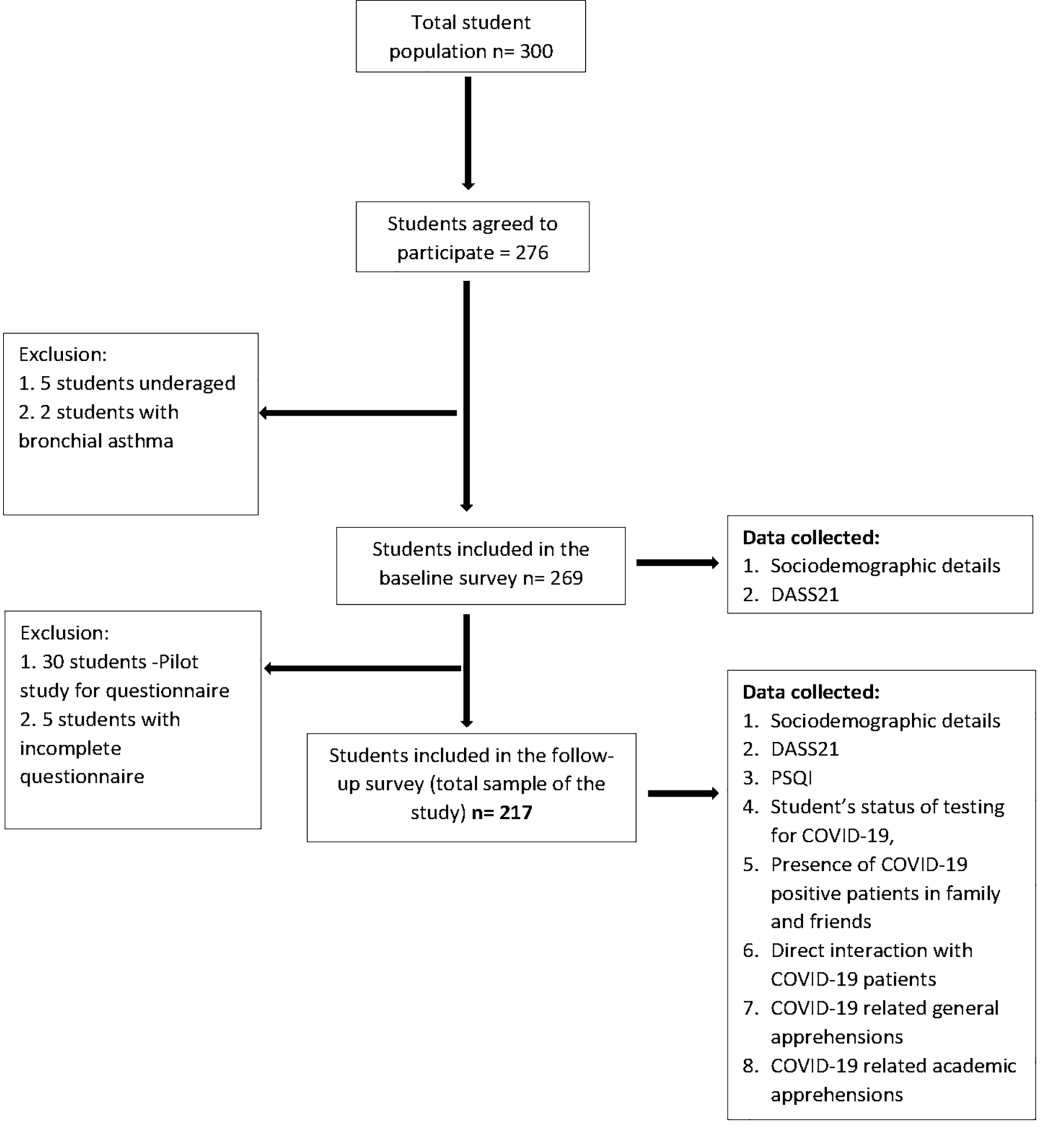

**Figure 1 Flow chart illustrating the sample selection from baseline to follow-up.** DASS21, Depression, Anxiety, Stress Scale 21 items; PSQI, Pittsburgh Sleep Quality Index.

empirically on 30 students (15 each from pre/para clinical years and clinical years) as a pilot study (*Hill, 1998*). The collected feedback and responses were analyzed, and corrections were made in the form of changes in articulation and simplification of vocabulary with the help of experts in this field.

## Identification of latent variables from the self-administered questionnaire

An exploratory factor analysis (EFA) was performed on the 9 items (item 4–12) to determine the validity of the questionnaire and to identify latent variables that could

enable the objective of the study. The EFA was conducted using principal component analysis with varimax rotation for factor extraction.

The extracted factors were analyzed for retention using Kaiser criterion (Eigen value >1), Scree test and counter-validated using parallel analysis. The Eigen values obtained from parallel analysis, which are values generated randomly with the same number of variables and sample size, were compared with the factor solution generated by EFA with our data. Eigen values of the factors that were higher than the values obtained from parallel analysis decided the number of factors. The two-factor solution thus obtained had 5 items in one (items 4–8) and 4 items in another (items 9–12). The factors were named COVID-19-related general apprehensions (COVID-19-GA) and COVID-19-related academic apprehensions (COVID-19-AA) respectively (Table S1). Reliability analysis was performed for each of the factors separately which presented with a high reliability, with the Cronbach's alpha score of 0.89 for COVID-19-GA and 0.91 for COVID-19-AA. COVID-19-GA was scored by totaling the scores of the five individual items with the total score ranging from 5 to 25 and higher scores denote higher general apprehension. Similarly, COVID-19-AA was scored by summing up the scores of the four individual items with the scores ranging from 4 to 20 and higher scores denote higher academic related apprehension.

## Estimation of mental health status

Mental health status of the medical students was assessed using Lovibond and Lovibond's DASS21 (*Lovibond & Lovibond, 1995*). This scale comprises of 21 items with seven each for depression, anxiety, and stress subscales. The total sub scores range from 0 to 42 and is categorized into normal, mild, moderate, severe, and extremely severe. In this study, DASS21 sub scores were categorized dichotomously, with the participants being divided in to those who showed symptoms of depression, anxiety and stress and those who did not, based on the cut-off sub-scores of 9, 7 and 14 respectively (*Cheung & Yip, 2015*; *Lovibond & Lovibond, 1995*).

## Estimation of sleep quality

Subjective sleep quality was assessed using PSQI which includes 21 items that assess seven components viz. subjective sleep quality, sleep duration, sleep latency, habitual sleep efficiency, use of sleep medications, sleep disturbance, and daytime dysfunction over the duration of 2 weeks prior to assessment. Global PSQI scores are obtained by summing up the seven individual sub scores and it ranges from 1 to 21 with higher scores (>5) denoting poor sleep quality (*Buysse et al., 1989*; *Rahe et al., 2015*).

Previous studies have shown high reliability of both DASS21 and PSQI among Indian undergraduate medical student population (*Shad, Thawani & Goel, 2015*; *Yadav, Gupta & Malhotra, 2016*). In our study, both the scales showed good internal consistency and DASS21 scale demonstrated good test-retest reliability. Cronbach alpha score for reliability for PSQI was 0.72 and for DASS21 0.94 (depression subscale 0.85, anxiety subscale 0.84, stress subscale 0.87) and 0.94 (depression subscale 0.87, anxiety subscale 0.81, stress subscale 0.85) for baseline and follow-up survey, respectively.

## Ethical consideration

Ethical approval was obtained from the IEC, Madha Medical College and Research Institute in Chennai (MMCRI/IEC/H/018/2020) and research was done in accordance with the Helsinki Declaration for research on human participants.

## Statistical analysis

Descriptive statistics was performed for all the variables. The scores of depression, anxiety, stress, and sleep quality were expressed as mean ± standard deviation (SD). Initially, unadjusted univariate association between the demographic variables and depression, anxiety and stress were performed. Mann Whitney U test and Kruskal Wallis test for continuous variables and Chi-square test for categorical variables were used for cross-sectional analysis. Wilcoxon signed rank test for continuous variables and McNemar's test for categorical variables were used for longitudinal analysis. Spearman's correlation test was performed to assess the correlation between the scores obtained from the survey instruments in both the surveys.

To explore the contributory factors associated with depression, anxiety, and stress during COVID-19 outbreak (dependent variable), adjusted binary logistic regression analysis was performed. Independent variables included were scores of PSQI, COVID-19-GA, COVID-19-AA, dependent variable from baseline survey (depression, anxiety and stress sub-scores in respective regression models) as covariates (continuous variables) and responses for the items 1–3 from the self-administered questionnaire as independent factors (categorical variables). Cross-sectional association between sleep quality and study parameters were analyzed using adjusted binary logistic regression. The effect of each of the independent variable was adjusted for sociodemographic variables which were considered to be potential confounders viz. age, gender, year of study, current residence, and family monthly income, in separate binary regression models. The results were expressed as adjusted odds ratio (aOR), 95% confidence interval (95% CI) and $P$ value (statistical significance set at two-tailed $P < 0.05$).

All the statistical analysis was performed using SPSS version 26 (IBM, NY, USA) and Parallel analysis was performed using scripts from *O'connor (2000)*.

# RESULTS

This longitudinal study includes 217 undergraduate medical students (78 males and 139 females); the average age was 20 ± 1.6 years. A total of 5.1% got tested for COVID-19, and they all tested negative. A total of 14.3% had friends and family who tested positive for COVID-19 and 12% declared to have had direct contact with COVID-19 patients (Fig. 2). The distribution of responses to the items of COVID-19-GA and COVID-19-AA is shown in Fig. 2 and percentage distribution of depression, anxiety and stress levels in baseline and follow-up survey and sleep quality in follow-up survey is shown in Fig. 3.

## Cross-sectional relationship between sociodemographic variables and depression, anxiety, and stress

The cross-sectional and longitudinal relationship between sociodemographic variables and depression, anxiety, and stress are shown in Table 1. There was no significant

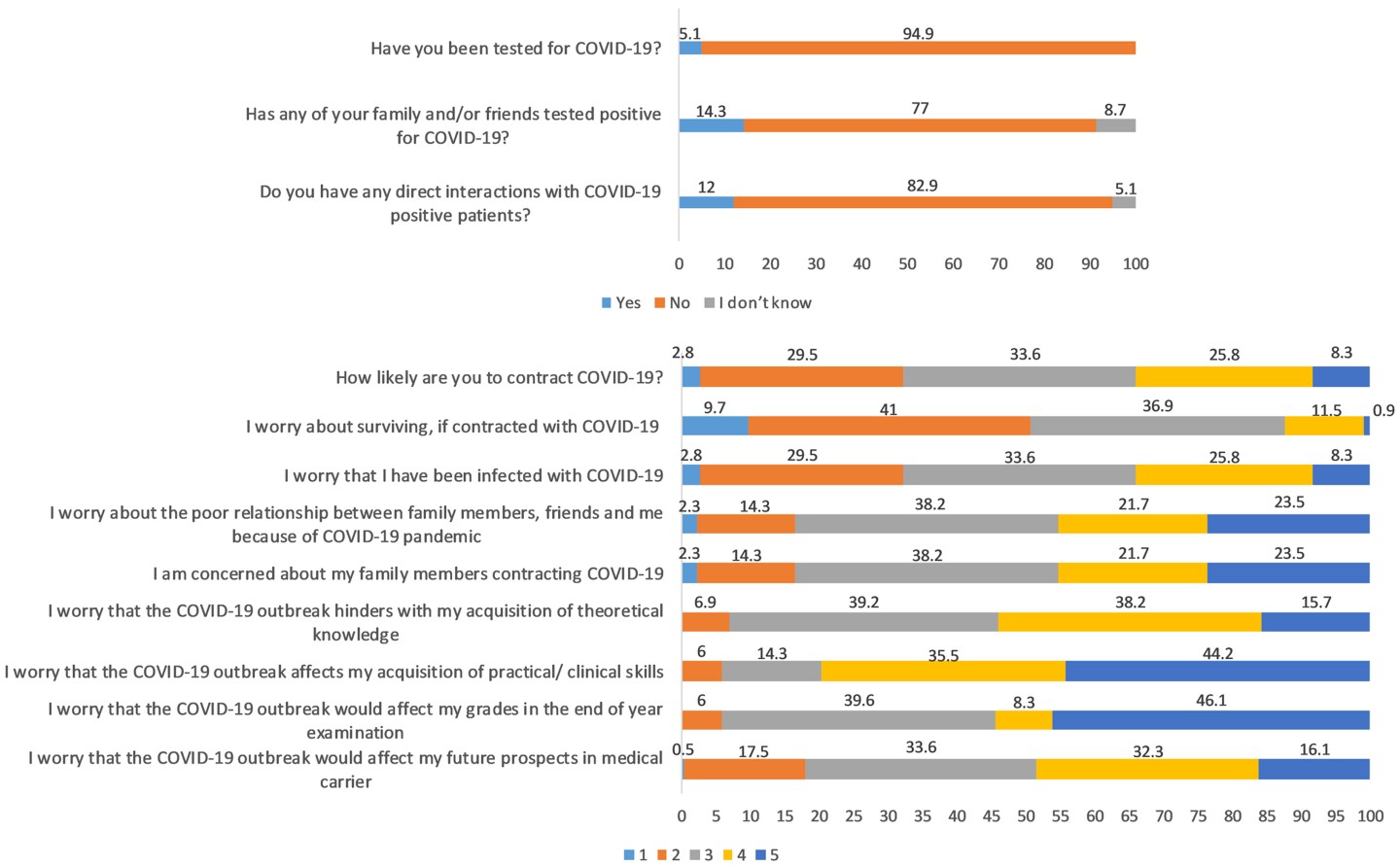

**Figure 2 Distribution of responses to self-administered questionnaire (in %).** Items 4 to 12: Likert scale score, 1 through 5; from minimum to maximum COVID-19-GA scores (5 to 25) = 4 + 5 + 6 + 7 + 8; COVID-19-AA scores (4 to 20) = 9 + 10 + 11 + 12.

cross-sectional relationship between the demographic variables and both baseline and follow-up depression, anxiety, and stress scores except in the baseline survey where depression levels were higher in the students from rural sector when compared to urban sector ($P = 0.039$). The association between demographic variables and depression, anxiety, and stress analyzed by binary logistic regression showed that age was a protective factor for depression in the follow-up survey (OR 0.737, 95% CI [0.565–0.961]) (Table S2). Other than this, there were no significant associations between demographic variables and mental health in both baseline and follow-up survey (Tables S2–S4).

## Comparison of baseline and follow-up depression, anxiety and stress stratified by sociodemographic variables

The overall prevalence (with 95% confidence interval) of depression, anxiety and stress before COVID-19 was 33.2% [27–39.9%], 21.2% [16–27.2%] and 20.7% [15.5–26.7%] and during COVID-19 outbreak was 35.5% [29.1–42.2%], 33.2% [27–39.9%] and 24.9% [19.3–31.2%] respectively. There was a significant increase in the prevalence and mean scores of anxiety and stress when compared to baseline scores ($P < 0.001$) (Table 1). In terms of prevalence, when compared to baseline values, the prevalence of anxiety

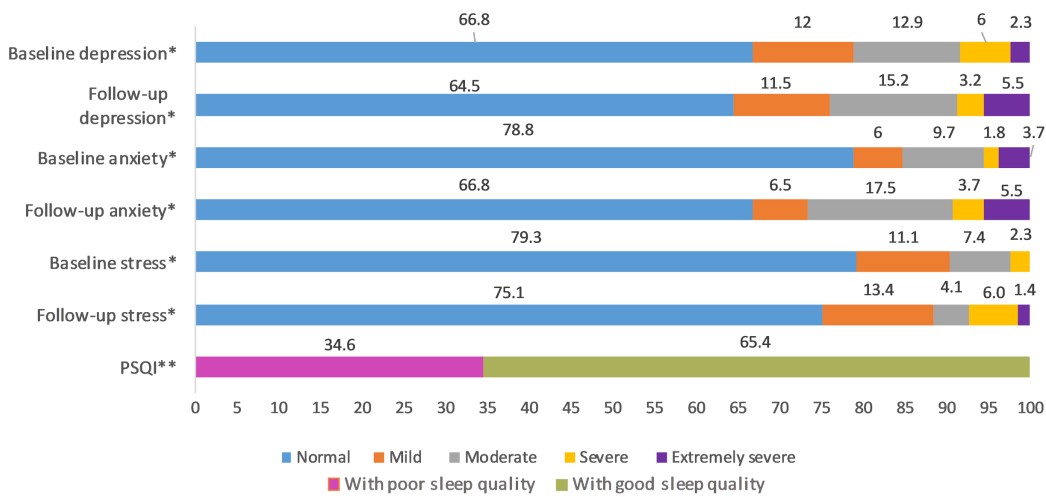

**Figure 3 Percentage distribution of baseline and follow-up depression, anxiety and stress levels and sleep quality in follow-up survey.** *Severity ranking scores DASS-42. Depression: normal; 0–9, mild; 10–13, moderate; 14–20, severe; 21–-27, extremely severe ≥ 28. Anxiety: normal; 0–7, mild; 8–9, moderate; 10–14, severe; 15–19, extremely severe ≥ 20. Stress: normal; 0–14, mild; 15–18, moderate; 19–25, severe; 26–33, extremely severe ≥ 34. DASS21 mean scores are doubled to calculate severity ranking (*Lovibond & Lovibond, 1995*). **Global PSQI score above five is considered to be poor sleep quality (*Buysse et al., 1989*).

was significantly increased during COVID-19 in females ($P$ = 0.003), pre-clinical, para-clinical ($P$ = 0.014) and clinical year students ($P$ = 0.019), students from urban residence ($P$ = 0.005) and those with gross family income above 100,000 INR per month ($P$ = 0.007). In terms of mean DASS21 scores, there was a statistically significant increase in anxiety with mild effect size and stress levels with moderate to strong effect size in medical students in the follow-up survey compared to baseline levels irrespective of gender, year of study, current residence and gross family income per month (below 50,000 INR and above 100,000 INR) ($P$ < 0.05). Out of the entire study population, the levels of depression have increased significantly in male students and students from urban residence when compared to before COVID-19 levels albeit with mild effect size (Table 1).

## Difference in ranks of DASS21 scores between baseline and follow-up survey

The difference in ranks of DASS21 scores between baseline and follow-up survey is shown in Table 2. There was no significant change in depression ($P$ = 0.146) in medical students between the two surveys; however, anxiety and stress has increased ($P$ < 0.001), showing an increase in median scores in the follow-up survey (Tables 1 and 2). The incidence of depression, anxiety, and stress based on DASS21 sub-scores were found to be 2.3 (5 out of 217), 11.98 (26 out of 217) and 4.15 (9 out of 217) per 100 per 6 months, respectively. 97 (44.7%), 89 (41.01%) and 142 (65.44%) students scored higher in depression, anxiety, and stress in the follow-up survey when compared to their responses before COVID-19 outbreak. While 60 (28.04%), 69 (31.79) and 7 (3.22%) students scored lesser than their baseline scores in depression, anxiety, and stress in follow-up survey (Table 2).
**Table 1  Relationship between demographic variables and baseline and follow-up depression, anxiety and stress scores.**

| Groups | | Depression | | | | Anxiety | | | | Stress | | | |
|---|---|---|---|---|---|---|---|---|---|---|---|---|---|
| | | Baseline | Follow-up | P value | Effect size (r) | Baseline | Follow-up | P value | Effect size (r) | Baseline | Follow-up | P value | Effect size (r) |
| Total sample (217) | Mean ± SD | 7.55 ± 7.86 | 8.16 ± 8.9 | 0.146[a] | −0.070 | 4.6 ± 6.19 | 6.11 ± 7.13 | **<0.001**[a] | −0.185 | 7.31 ± 7.34 | 9.31 ± 8.18 | **<0.001**[a] | −0.495 |
| | N (%) | 72 (33.2) | 77 (35.5) | 0.609[b] | | 46 (21.2) | 72 (33.2) | **<0.001**[b] | | 45 (20.7) | 54 (24.9) | **<0.001**[b] | |
| Gender Male (78) | Mean ± SD | 7.28 ± 8.4 | 8.54 ± 9.17 | **0.023**[a] | −0.181 | 4.62 ± 6.04 | 6.41 ± 7.5 | **0.010**[a] | −0.208 | 7.95 ± 7.54 | 10.08 ± 8.5 | **<0.001**[a] | −0.528 |
| | N (%) | 21 (26.9) | 29 (37.2) | 0.152[b] | | 20 (25.6) | 27 (34.6) | 0.118[b] | | 18 (23.1) | 23 (29.5) | 0.063[b] | |
| Female (139) | Mean ± SD | 7.71 ± 7.57 | 7.94 ± 8.77 | 0.953[a] | −0.004 | 4.59 ± 6.29 | 5.94 ± 6.93 | **0.004**[a] | −0.173 | 6.95 ± 7.22 | 8.88 ± 7.99 | **<0.001**[a] | −0.479 |
| | N (%) | 51 (36.7) | 48 (34.5) | 0.743[b] | | 26 (18.7) | 45 (32.4) | **0.003**[b] | | 27 (19.4) | 31 (22.3) | 0.344[b] | |
| P[c] | | 0.409 | 0.433 | | | 0.986 | 0.505 | | | 0.446 | 0.304 | | |
| P[d] | | 0.143 | 0.696 | | | 0.230 | 0.737 | | | 0.524 | 0.240 | | |
| Year of study Pre/paraclinical years (150) | Mean ± SD | 7.35 ± 7.77 | 7.84 ± 9.04 | 0.354[a] | −0.054 | 4.88 ± 6.13 | 6.31 ± 7.52 | **0.009**[a] | −0.151 | 7.08 ± 7.36 | 9.05 ± 8.29 | **<0.001**[a] | −0.496 |
| | N (%) | 50 (33.3) | 51 (34) | 1.000[b] | | 34 (22.7) | 49 (32.7) | **0.014**[b] | | 30 (20) | 35 (23.3) | 0.180[b] | |
| Clinical years (67) | Mean ± SD | 8 ± 8.11 | 8.87 ± 8.59 | 0.203[a] | −0.110 | 3.97 ± 6.32 | 5.67 ± 6.18 | **0.003**[a] | −0.253 | 7.82 ± 7.33 | 9.88 ± 7.95 | **<0.001**[a] | −0.495 |
| | N (%) | 22 (32.8) | 26 (38.8) | 0.503[b] | | 12 (17.9) | 23 (34.3) | **0.019**[b] | | 15 (22.4) | 19 (28.4) | 0.219[b] | |
| P[c] | | 0.621 | 0.266 | | | 0.079 | 0.759 | | | 0.350 | 0.339 | | |
| P[d] | | 0.943 | 0.494 | | | 0.428 | 0.810 | | | 0.689 | 0.429 | | |
| Current residence Rural (76) | Mean ± SD | 9.36 ± 9.38 | 8.71 ± 8.87 | 0.587[a] | 0.044 | 4.92 ± 6.54 | 6.66 ± 7.96 | **0.010**[a] | −0.210 | 8.05 ± 7.61 | 10.16 ± 8.58 | **<0.001**[a] | −0.473 |
| | N (%) | 28 (36.8) | 29 (38.2) | 1.000[b] | | 18 (23.7) | 26 (34.2) | 0.057[b] | | 19 (25) | 22 (28.9) | 0.453[b] | |
| Urban (141) | Mean ± SD | 6.58 ± 6.75 | 7.86 ± 8.93 | **0.022**[a] | −0.136 | 4.43 ± 6.01 | 5.82 ± 6.65 | **0.004**[a] | −0.171 | 6.91 ± 7.18 | 8.85 ± 7.95 | **<0.001**[a] | −0.509 |
| | N (%) | 44 (31.2) | 48 (34) | 0.627[b] | | 28 (19.9) | 46 (32.6) | **0.005**[b] | | 26 (18.4) | 32 (22.7) | 0.07[b] | |
| P[c] | | **0.039** | 0.244 | | | 0.806 | 0.769 | | | 0.245 | 0.297 | | |
| P[d] | | 0.400 | 0.546 | | | 0.511 | 0.813 | | | 0.255 | 0.310 | | |
| Family monthly income <50,000 INR (72) | Mean ± SD | 9.67 ± 9.73 | 9.78 ± 10.45 | 0.696[a] | −0.033 | 5.08 ± 7.14 | 7.06 ± 8.36 | **0.023**[a] | −0.190 | 8.50 ± 7.86 | 10.86 ± 8.97 | **<0.001**[a] | −0.513 |
| | N (%) | 31 (43.1) | 29 (40.3) | 0.824[b] | | 19 (25.6) | 26 (33.3) | 0.210[b] | | 18 (25) | 24 (33.3) | 0.07[b] | |
| 50,000–100,000 INR (84) | Mean ± SD | 6.83 ± 7.01 | 7.86 ± 8.62 | 0.194[a] | −0.100 | 5.17 ± 6.74 | 5.88 ± 6.7 | 0.193[a] | −0.100 | 6.88 ± 7.05 | 8.81 ± 7.92 | **<0.001**[a] | −0.492 |
| | N (%) | 26 (32.1) | 29 (37.2) | 0.481[b] | | 18 (23.1) | 26 (33.3) | 0.134[b] | | 15 (17.9) | 15 (17.9) | 1.000[b] | |
| >100,000 INR (61) | Mean ± SD | 6.05 ± 5.84 | 6.66 ± 6.88 | 0.471[a] | −0.065 | 3.25 ± 3.47 | 5.31 ± 6.02 | **0.001**[a] | −0.298 | 6.49 ± 7.02 | 8.16 ± 7.38 | **<0.001**[a] | −0.481 |
| | N (%) | 15 (24.6) | 19 (31.1) | 0.523[b] | | 9 (14.8) | 20 (32.8) | **0.007**[b] | | 12 (19.7) | 15 (24.6) | 0.250[b] | |
| P[e] | | 0.175 | 0.375 | | | 0.383 | 0.616 | | | 0.229 | 0.193 | | |
| P[d] | | 0.068 | 0.533 | | | 0.262 | 0.790 | | | 0.532 | 0.084 | | |

**Notes:**
[a] P value: Wilcoxon signed rank test.
[b] P value: McNemar test.
[c] P value: Mann Whitney U test.
[d] P value: Chi square test.
[e] P value: Kruskal Wallis test.
r value: effect size of Wilcoxon signed rank test.
N (%): Number of subjects with depression, anxiety and stress.
Subjects with depression: depression sub-score > 9.
Subjects with anxiety: anxiety sub-score > 7.
Subjects with stress: stress sub-score > 14.
Significant P values are shown in bold.

**Table 2 Difference in the ranks of DASS21 scores between baseline and follow-up surveys.**

| Score difference | Negative ranks[a] (N) | Positive ranks[b] (N) | Ties[c] (N) | Median score (50th percentile) | | Wilcoxon signed-rank test P-value |
|---|---|---|---|---|---|---|
| | | | | Baseline | Follow-up | |
| Depression | 60 | 97 | 60 | 6 | 6 | 0.146 |
| Anxiety | 69 | 89 | 59 | 2 | 4 | **<0.001** |
| Stress | 7 | 142 | 68 | 6 | 8 | **<0.001** |

Notes:
[a] follow-up < baseline.
[b] follow-up > baseline.
[c] follow-up = baseline.
Significant P values are shown in bold.

**Table 3 Correlation between scores of survey instruments from baseline and follow-up survey.**

| Variables | COVID-19-AA | COVID-19-GA | PSQI | Depression (baseline) | Depression (follow-up) | Anxiety (baseline) | Anxiety (follow-up) | Stress (baseline) | Stress (follow-up) |
|---|---|---|---|---|---|---|---|---|---|
| COVID-19-AA | 1.000 | – | – | – | – | – | – | – | – |
| COVID-19-GA | -0.026 (0.703) | 1.000 | – | – | – | – | – | – | – |
| PSQI | 0.030 (0.664) | 0.032 (0.638) | 1.000 | – | – | – | – | – | – |
| Depression (baseline) | −0.025 (0.710) | **0.152*** **(0.025)** | **0.422**** (<0.001) | 1.000 | – | – | – | – | – |
| Depression (follow-up) | 0.019 (0.782) | 0.072 (0.292) | **0.520**** (<0.001) | **0.673**** (<0.001) | 1.000 | – | – | – | – |
| Anxiety (baseline) | −0.010 (0.886) | 0.126 (0.063) | **0.296**** (<0.001) | **0.566**** (<0.001) | **0.511**** (<0.001) | 1.000 | – | – | – |
| Anxiety (follow-up) | 0.012 (0.857) | 0.088 (0.199) | **0.389**** (<0.001) | **0.594**** (<0.001) | **0.712**** (<0.001) | **0.502**** (<0.001) | 1.000 | – | – |
| Stress (baseline) | 0.011 (0.873) | 0.130 (0.055) | **0.474**** (<0.001) | **0.710**** (<0.001) | **0.783**** (<0.001) | **0.575**** (<0.001) | **0.765**** (<0.001) | 1.000 | – |
| Stress (follow-up) | −0.007 (0.914) | 0.130 (0.056) | **0.460**** (<0.001) | **0.690**** (<0.001) | **0.771**** (<0.001) | **0.571**** (<0.001) | **0.729**** (<0.001) | **0.961**** (<0.001) | 1.000 |

Notes:
* $P < 0.05$.
** $P < 0.001$.
The results are expressed as ρ (Rho) value.
Significant P values are shown in bold.

## Correlation between the scores of the survey instruments from baseline and follow-up survey

The results of Spearman correlation analysis of the scores of the survey instruments from baseline and follow-up survey is shown in Table 3. There were significant positive correlations between PSQI and baseline and follow-up depression, anxiety, and stress. The correlation between the baseline and follow-up depression, anxiety, and stress scores indicates that a higher baseline score was associated with higher follow-up score and vice versa ($P < 0.001$). There was a relatively weak, but significant positive correlation between depression levels before COVID-19 and COVID-19-related general apprehensions ($r = 0.152$, $P = 0.025$) (Table 3).

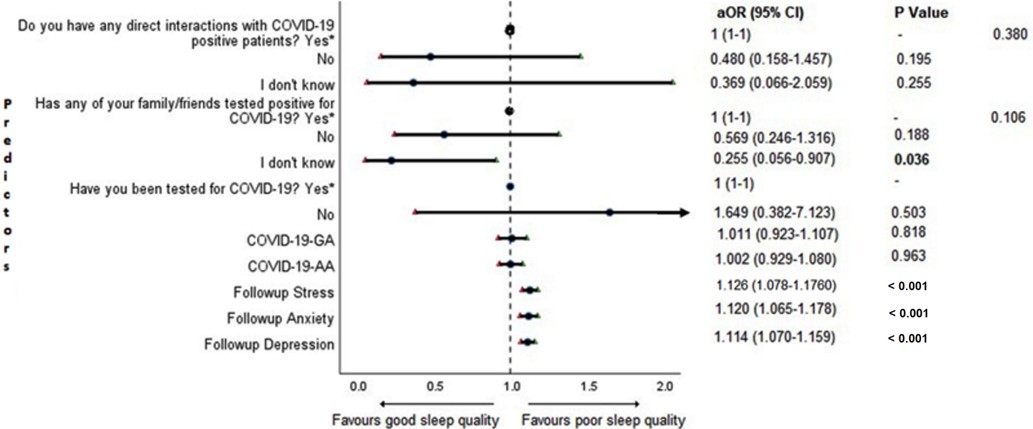

**Figure 4 Forest plot showing adjusted binary logistic regression analysis of follow-up PSQI scores (cross sectional).** aOR adjusted odds ratio; odds ratio adjusted for age, gender, year of study, urban/ rural residential status, family's monthly income; 95%CI 95% confidence interval. Significant *P* values are shown in bold.

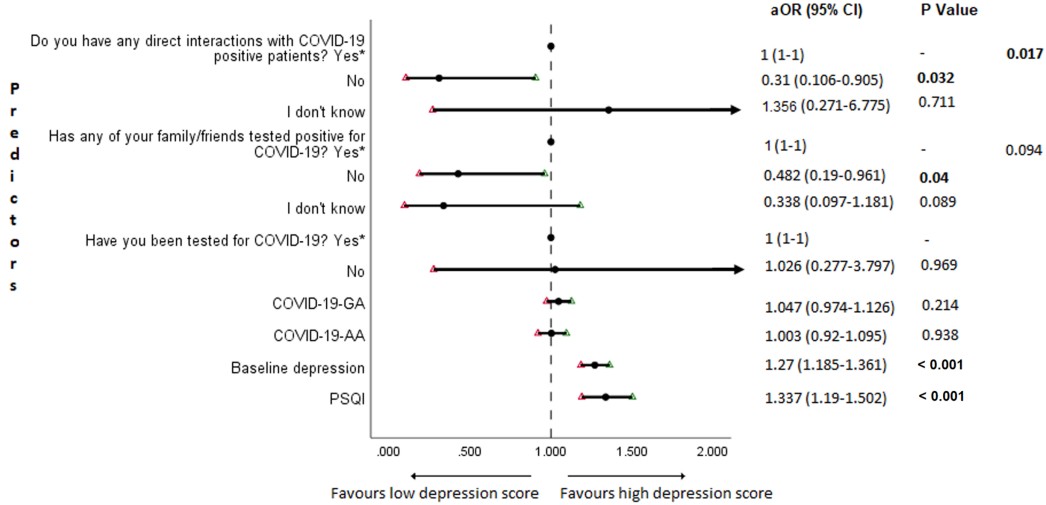

**Figure 5 Forest plot showing adjusted binary logistic regression analysis of follow-up depression scores.** aOR adjusted odds ratio; odds ratio adjusted for age, gender, year of study, urban/rural residential status, family's monthly income; 95%CI 95% confidence interval. Significant *P* values are shown in bold.

## Adjusted binary logistic regression analysis of sleep quality

Cross-sectional association between sleep quality and mental health in the follow-up survey is shown in Fig. 4. Students with higher depression, anxiety, and stress scores during COVID-19 outbreak were found to be more likely to have poor sleep quality (*P* < 0.001).

## Adjusted binary logistic regression analysis of follow-up depression, anxiety, and stress

The results of binary logistic regression analysis for follow-up depression, anxiety, and stress are shown in Figs. 5–7. Poor sleep quality was found to be significantly associated

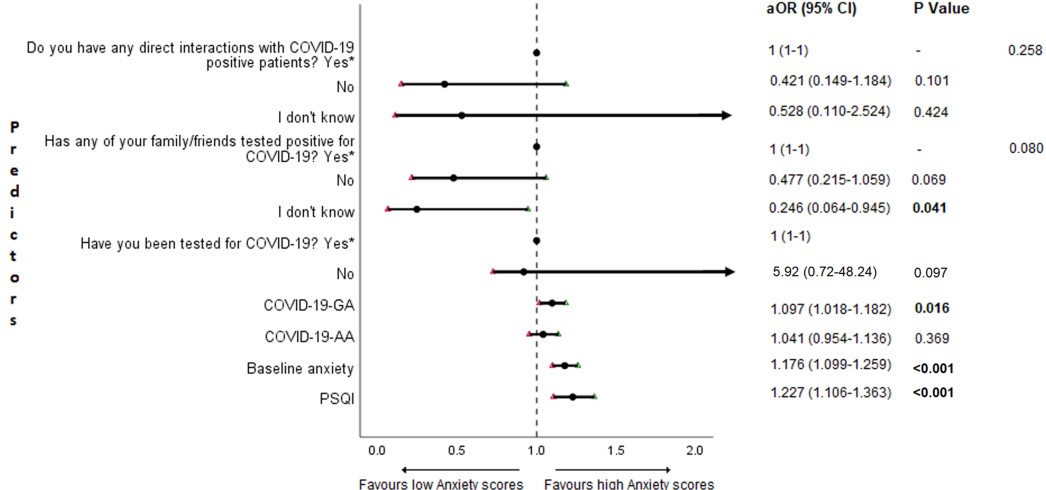

**Figure 6 Forest plot showing adjusted binary logistic regression analysis of follow-up anxiety scores.** aOR adjusted odds ratio; odds ratio adjusted for age, gender, year of study, urban/rural residential status, family's monthly income; 95%CI 95% confidence interval. Significant *P* values are shown in bold.

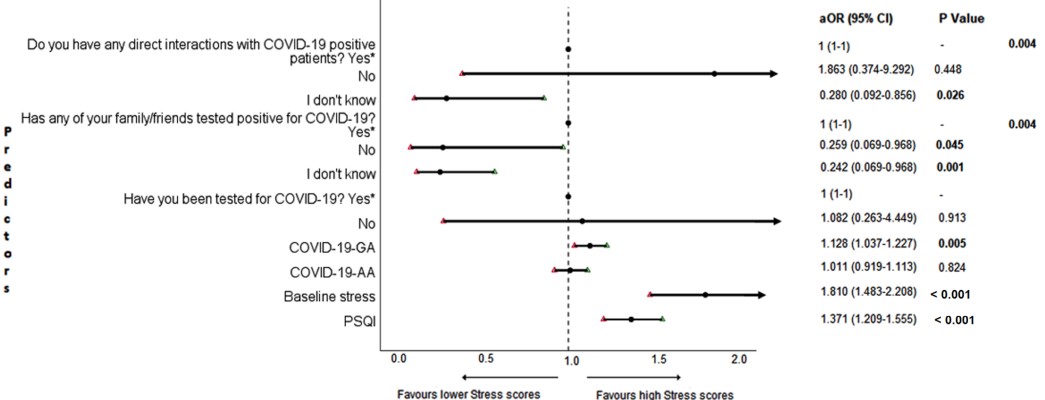

**Figure 7 Forest plot showing adjusted binary logistic regression analysis of follow-up stress scores.** aOR adjusted odds ratio; odds ratio adjusted for age, gender, year of study, urban/rural residential status, family's monthly income; 95%CI 95% confidence interval. Significant *P* values are shown in bold.

with an increase in depression, anxiety, and stress ($P < 0.001$). Higher baseline scores of depression, anxiety, and stress were associated with higher levels of the same in follow-up survey ($P < 0.001$). Higher COVID-19-related general apprehension was associated with higher levels of anxiety ($P = 0.016$), and stress ($P = 0.005$). Students who did not have any direct interactions with COVID-19 patients were found to be less likely to have symptoms of depression ($P = 0.017$) and stress ($P = 0.004$) when compared to those who did. Similarly, absence of COVID-19 patients in family and friends was found to be associated with decreased levels of stress ($P = 0.004$).

## DISCUSSION

The present study investigated the mental health status of undergraduate medical students in a medical college, which is a government-approved center for treating COVID-19 patients. The medical college is in Chennai, TamilNadu, which is one of the top 5 COVID-19 affected metropolitan cities in India. Longitudinal data analysis was used to test our hypothesis that COVID-19 outbreak and quarantine has a negative impact on the mental health of undergraduate medical students.

We found that 35.5% (95% CI [29.1–42.2%]), 33.2% (95% CI [27–39.9%]) and 24.9% (95% CI [19.3–31.2%]) of the undergraduate medical students, including resident interns showed symptoms of depression, anxiety, and stress respectively during COVID-19 outbreak with the majority with moderate depression (15.2%), moderate anxiety (17.5%), and mild stress (13.4%). Based on the severity ranking, subjects with moderate and above ranking may present with a possible problem that may require intervention (*Crawford & Henry, 2003*; *Lovibond & Lovibond, 1995*; *Page, Hooke & Morrison, 2007*; *Stormon et al., 2019*). In this study, 23.9%, 26.7% and 11.5% of the study population presented with moderate to extremely severe levels of depression, anxiety, and stress respectively (Fig. 3). The 6-month incidence of anxiety was found to be comparatively higher (11.98%) followed by stress (4.15%) and depression (2.3%). There were no other longitudinal studies conducted in medical students during COVID-19 pandemic assessing the incidence of mental health disorders; however, a study conducted in the general public in China found the 1-month incidence of mental health disorders to be relatively low (*Ren et al., 2020*). Though the incidence rate seemed to be relatively low in our study, compared to baseline levels, 44.7%, 41.01% and 65.44% of the study population scored higher in depression, anxiety, and stress sub scores during COVID-19 (Table 2). When compared to baseline survey that was recorded before COVID-19 outbreak in India, there was a significant increase in prevalence and levels of anxiety and stress, while that of the depression remained unchanged during COVID-19. Similar studies conducted longitudinally in college students found a significant increase in depression and anxiety when compared to before COVID-19 levels (*Li et al., 2020a*, *Huckins et al., 2020*). This negative impact of the pandemic could be attributed to sudden challenges faced by the medical students in terms of academics, uncertainties about future, fear of infection, news about shortage of personal protective equipment, quarantine induced boredom, frustrations, lack of freedom, and fears caused by rumors and misleading news in the media (*Bao et al., 2020*; *Ferrel & Ryan, 2020*).

Investigation of the influence of demographics on mental health showed that the increase in anxiety and stress levels in our study population was not affected by gender, year of study, current residence, or family monthly income. We also found no significant cross-sectional difference in depression, anxiety, and stress between the groups of demographic variables during COVID-19. Binary logistic regression analysis of anxiety and stress and demographic variables as independent variables showed no significant association. In contrary, cross-sectional studies done on medical students in China and Brazil during this pandemic found significant associations between mental health

disorders and place of residence, parental income/financial support, and gender (*Cao et al., 2020*; *Sartorao Filho et al., 2020*). Our results show that the response of the medical students to COVID-19 pandemic in terms of the levels of anxiety and stress is similar, irrespective of gender, year of study, current residence, and financial status of the family. Depression in the study population, on the other hand, remained unchanged during COVID-19 in all the categories except in male and urban population. Binary logistic regression showed that increase in age could decrease the likelihood of depression (OR 0.737, 95% CI [0.565–0.961]) which is consistent to previous studies during COVID-19 (*Ahmed et al., 2020a*; *González-Sanguino et al., 2020*). However, in our study, the range of age was narrow (18–26 years) and mostly corresponded to the year of study, which had no significant association with depression. Hence, this observation becomes redundant. We found a significant increase in depression levels in male students but not in female students. This finding is contradictory to the study by *Sartorao Filho et al. (2020)* which found female medical students to be more at risk of developing depression symptoms during this pandemic. Females are more proactive in their response and awareness about the epidemic when compared to males, and our results could be a possible implication of this (*Brittni Frederiksen, Salganicoff & Ranji, 2020*). Despite higher levels of depression in the students from rural areas before COVID-19, their depression levels remained unchanged in the follow-up survey. However, the students from the urban areas presented with an increase in depression when compared to before COVID-19 levels. Nearly 53% of India's cases were recorded in Mumbai, Delhi, Ahmedabad, Pune, and Chennai alone, which were listed as top five COVID-19 cities (*Shylendra, 2020*). Urban areas are highly populated, and the epidemic is more active in these areas than in rural areas. This led to the implementation of frequent lockdown measures by the respective State Governments in the COVID-19 hotspots, which are mainly urban centers. This could have possibly increased the psychological distress in the students from urban locations.

Since we did not find significant differences in mental health among most of the groups of demographic variables in both baseline and follow-up survey, it is likely that the worsening of mental health status of the medical students found in our study is associated with COVID-19-related factors. To further elucidate this, we used adjusted binary logistic regression analysis to explore possible predictors. Our extensive literature survey showed all the collected demographic variables to have a potential influence on the outcome. Hence, despite not finding significant associations between demographics and mental health, we adjusted the effects of each potential predictor for all the recorded demographic variables in the regression models (*Cao et al., 2020*; *Sartorao Filho et al., 2020*; *Heinze & Dunkler, 2017*).

An important finding in our study is the independent bidirectional association between poor sleep quality and mental health. In our study, 34.6% of the study population suffered from poor sleep quality which was found to be a significant independent predictor of depression (aOR 1.337, 95% CI [1.19–1.502]), anxiety (aOR 1.227, 95% CI [1.106–1.363]) and stress (aOR 1.371, 95% CI [1.209–1.555]) during COVID-19 (Figs. 5–7), similar to a previous study by *Cellini et al. (2020)* in Italy. Medical students are especially prone to poor sleep quality because of the physically and emotionally

challenging and intense training they undertake (*Wong et al., 2005*). Poor sleep affects neurocognitive and psychomotor performance, emotional wellbeing, working capacity, academic performance, physical and mental health as well as quality of life (*Al-Khani et al., 2019*; *Flores, 2009*; *Giri, Baviskar & Phalke, 2013*; *Mume, Olawale & Osundina, 2011*). Due to lockdown measures and travel restrictions, students are facing decreased physical activity, lack of schedule, altered living conditions, increased screen time and time spent in social media, and altered sleep wake schedule including increased daytime nap duration (*Majumdar, Biswas & Sahu, 2020*). All these factors in addition to higher demands of medical curriculum could lead to poor sleep, which in turn affects mental wellbeing. Conversely, we also found that higher depression (aOR 1.114, 95% CI [1.07–1.159]), anxiety (aOR 1.120, 95% CI [1.065–1.178]) and stress levels (aOR 1.126, 95% CI [1.078–1.176]) during COVID-19 were significant predictors of poor sleep quality (Fig. 4). Poor sleep is long since been considered an important symptom of mental health disorders. Sleep disturbance is a primary symptom of major depressive disorder (*Jindal & Thase, 2004*). Anxiety and stress negatively affect the body's ability to fall and stay asleep (*Coplan et al., 2015*; *Kalmbach, Anderson & Drake, 2018*). In addition, increased time spent in social media and digital devices is a way by which young adults cope with social isolation, and it is associated with increased tendencies to develop sleep disturbances (*Sivertsen et al., 2019*). Thus, our findings show that poor sleep quality is both a cause and an effect of increased depression, anxiety, and stress symptoms in medical students during this pandemic. Thus, worsening of one could exacerbate the other.

Students with higher baseline levels of depression, anxiety, and stress were found to be more likely to have depression (aOR 1.27, 95% CI [1.185–1.361]), anxiety (aOR 1.176, 95% CI [1.099–1.259]) and stress (aOR 1.810, 95% CI [1.483–2.208]) during COVID-19 outbreak (Figs. 5–7). There was also significant positive correlation between baseline and follow-up depression, anxiety, and stress scores (Table 3). These results are consistent with a similar study conducted on college students in China (*Li et al., 2020a*). Studies show associations between pre-existing mental health problems and mental health disorders in medical students (*Yates, James & Aston, 2008*). Previous studies indicate that COVID-19 has a higher negative impact on people with mental health disorders when compared to those without any (*Asmundson et al., 2020*; *Taylor et al., 2020*). Medical students are proven to be at a higher risk of developing mental health disorders during the course of their training period (*Moffat et al., 2004*). With the added stressors related to the pandemic, it might be difficult to cope, especially for those students who had higher levels of depression, anxiety, and stress to begin with, leading to exacerbation of symptoms.

A surprising finding in our study was that COVID-19-related academic apprehensions were not significantly associated with depression, anxiety, and stress. This is in contrary to a recent study conducted in medical students in China, which found moderate positive correlation between worries about academic delay and anxiety (*Cao et al., 2020*). Although social desirability response bias could be an attributing factor, our finding could be a reflection of the feel of assurance by the students because of drastic student-centered efforts taken by the medical college and universities in continuing medical education.

Another reason could be that, since these lockdown instigated changes were in effect for around 3 months, the students would have adapted to this new normalcy in their training better than anticipated. On the other hand, higher COVID-19-GA scores were found to be a significant predictor for higher levels of anxiety (aOR 1.097, 95% CI [1.018–1.182]) and stress (aOR 1.128, 95% CI [1.037–1.227]) (Figs. 6 and 7). In a study conducted on medical students in Pakistan, around 76% of the participants conveyed being worried about contracting COVID-19 during clinical postings and even more worried about insufficient care and improper treatment, if they contracted the infection (*Ahmed et al., 2020b*). In a recent study conducted on adult Indian population, individuals with increased self-perceived risk of contracting COVID-19 were found to be more likely to have mental health disorders (*Saikarthik, Saraswathi & Siva, 2020*). A longitudinal study conducted in China in college students, found fear of infection to be significantly associated with anxiety and depression (*Li et al., 2020a*). Our results demonstrated that medical students' self-perceived levels of worries for the self, family, and friends about contracting COVID-19; surviving if contracted with COVID-19; and COVID-19 affecting interpersonal relationships have a negative impact on their mental health. Student population is highly active in social media, which is filled with high amounts of misinformation, adding fear, and affecting mental well-being. Frequent use of social media was associated with higher prevalence of mental health problems during COVID-19 (*Gao et al., 2020*). At the same time, social media could be used for communications among peers and family, thereby offering much needed social support.

In this study, we found no significant association between being tested for COVID-19 and depression, anxiety, and stress. The possible reason could be that all of those who got tested for COVID-19 were found negative. We also found that students without any COVID-19 patients among their family and friends (aOR 0.259, 95% CI [0.069–0.968]) and those who were not sure about having any direct interactions with COVID-19 patients (aOR 0.280 95% CI [0.092–0.856]) were found to be less likely to have symptoms of stress (Fig. 7). Students without any direct interactions with COVID-19 patients were less likely to have symptoms of depression (aOR 0.31 95% CI [0.106–0.905]) (Fig. 5) when compared to those who did. These findings were in line with the study by *Cao et al. (2020)* in China. The R naught (R0) of SARS-CoV-2 is 2.2 (2-2.5), that is, each infected person spreads the infection to 2.2 individuals, in other words it is more contagious than seasonal flu (*Li et al., 2020b*). This high contagious nature of the novel SARS-CoV-2 virus could be related to our findings.

## Strengths and limitations of the study

The longitudinal nature of this study is its major strength which outweighs the limitation of relatively small sample size. The study investigated the mental health status of the same medical students before and during COVID-19 pandemic, which enabled in studying the pattern, temporal order and predictors of changes in their mental health status. To our knowledge, this is the first study to analyze the effects of COVID-19 outbreak on the mental health of undergraduate medical students longitudinally. We elaborately studied the influence of demographic variables, stressors related to COVID-19 by categorizing

them into general and academic apprehensions as well as sleep quality, on mental health. With the known impact of negative mental health on medical students' career and life, we believe the results of our study is important due to the insights provided, that will help the medical educators to address and devise strategies to overcome the pandemic-induced negative impact on undergraduate medical students' mental health.

Our study is not without limitations. Though our study population includes students who were tested for COVID-19, none turned out positive. Thus, our results cannot be extrapolated to medical students infected with the virus. Despite promising confidentiality, there could have been possible response bias by the students in answering the survey.

## CONCLUSIONS

Prevalence and levels of anxiety and stress increased, and depression symptoms remained unaltered during COVID-19 outbreak and quarantine. Poor sleep quality, higher levels of depression, anxiety, and stress before COVID-19, increased worries for the self, family, and friends about contracting and surviving, if contracted with COVID-19, and about COVID-19 affecting interpersonal relationships, presence of COVID-19 patients in family and friends and direct interactions with COVID-19 patients were found to be significant predictors of negative mental health in medical students.

Neglecting the mental health of the medical students would lead to long-term detrimental effects, which not only will affect the quality of life of medical students and future physicians, but also the overall performance of the healthcare system. An effective plan to safeguard the mental health of this already vulnerable population of undergraduate medical students is crucial. We strongly believe our findings would help the medical educators in addressing and mitigating the rise in mental health disorders, which could prove worse than the current pandemic itself. Further studies to analyze the temporal pattern of changes in mental health status of the medical students are warranted.

## ACKNOWLEDGEMENTS

The authors would like to thank the Deanship of Scientific Research, Majmaah University, Kingdom of Saudi Arabia. The authors would also like to thank all the students for actively taking part in this study.

### Funding

The authors received no funding for this work.

### Competing Interests

The authors declare that they have no competing interests.

### Author Contributions

- Ilango Saraswathi conceived and designed the experiments, analyzed the data, prepared figures and/or tables, and approved the final draft.

● Jayakumar Saikarthik conceived and designed the experiments, analyzed the data, prepared figures and/or tables, and approved the final draft.
● K. Senthil Kumar conceived and designed the experiments, performed the experiments, prepared figures and/or tables, and approved the final draft.
● Kumar Madhan Srinivasan performed the experiments, prepared figures and/or tables, authored or reviewed drafts of the paper, and approved the final draft.
● M. Ardhanaari performed the experiments, authored or reviewed drafts of the paper, and approved the final draft.
● Raghunath Gunapriya performed the experiments, authored or reviewed drafts of the paper, and approved the final draft.

## Human Ethics

The following information was supplied relating to ethical approvals (i.e., approving body and any reference numbers):

Madha Medical College and Research Institute granted Ethical approval to carry out the study within its facilities (MMCRI/IEC/H/018/2020).

## Data Availability

The raw data are available in a Supplemental File.

## Supplemental Information

Supplemental information for this article can be found online at http://dx.doi.org/10.7717/peerj.10164#supplemental-information.

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
