# Peer review of "Impact of COVID-19 outbreak on the mental health status of undergraduate medical students in a COVID-19 treating medical college: a prospective longitudinal study"

_PeerJ, doi:10.7717/peerj.10164_

## Round 0.1 · original submission · Major Revisions

Thank you for submitting this interesting study, however, as suggested by our reviewers, the paper still has major issues. Please address the paper accordingly.

·

Basic reporting

Overall the manuscript has been prepared in clear English with a good structure including appropriate use of figures and tables; literature review seemed to be adequate, and relevant results presented.
There are 3 comments for the authors to consider revising:
• The numbering unit of ‘lakh’ is not an internationally well-understood concept, could the authors please consider revising this in Introduction.
• It’s not very clear if there have been other longitudinal studies done before, on the effect of COVID on depression, anxiety, and stress in other populations/contexts. Please add something about this in the Introduction.
• Figures 1 and 2 appear to be in low-resolution.

Experimental design

The study appears to be original research with well-defined research question. The Introduction highlighted the need for this study/more research on the impact of COVID-19 on high-risk populations such as medical students. The methodological design seemed to be adequate to address the research questions, and is described with sufficient details.

Validity of the findings

Although the study is based on a somewhat small sample size, the spontaneity of the study (or outbreak of the pandemic) and the longitudinal nature of the data would out-weigh this issue. Statistical analysis appears to be appropriate and thorough.
There are 2 comments for the authors to consider revising:
• For Results, please consider adding effect sizes where-ever possible, especially to the changes in depression, anxiety and stress from baseline to follow-up survey, regardless of whether the difference was p < .05 or not.
• Line 355 – I believe ‘a relatively weak but significant positive correlation’ is more accurate than ‘a mild significant positive correlation’.
• For Discussion, please talk about, whether there has been other longitudinal studies conducted on the change of depression, anxiety, and stress from before to during COVID, and if the magnitude of change found in the current study is similar/different to that found for other populations in the other studies.

Additional comments

Some typos noted:
• Line 221 – literature search
• Line 349 – the full stop should be placed after (table 2).
• Line 450 – missing “and” in sentence: challenging and intense training

Reviewer 2 ·

Basic reporting

This study investigated the mental health of medical students in India. The paper was well organized and written. The research background was provided sufficiently. Raw data was shared. The major strengthening of the study was longitudinal study-design and validation of questionnaire for the COVID-19 related data. However, several concerns should be addressed.

Experimental design

Several details should be added, including the study site, sampling method at baseline, and the dependent and independent variables in the adjusted binary logistic regression. The limitations of small sample size should be addressed.

Validity of the findings

The results were provided sufficiently. However, the results should be separated into sub-sectionals for better understanding. The titles of all tables (including the supplementary tables) should be added. The statistic values should be added in both the results section and all tables. The P=0.000 should be written as P<0.001 instead.

Additional comments

No other comments.

Reviewer 3 ·

Basic reporting

None

Experimental design

None

Validity of the findings

None

Additional comments

This is a prospective longitudinal study, which aimed to investigate the mental health of undergraduate medical students over a duration of 6 months by analyzing data collected before and during COVID-19 outbreak in India. However, I have some comments for the authors for their consideration.
Abstract
1. Authors should report 95%CI of prevalence.
2. DASS 21 should report the full name when authors first mentioned it.
3. Authors should report specific statistic values about predictors (e.g., OR, 95%CI.)
Method
1. Authors should draw a flowchart to illustrate the sample selection from baseline to follow-up.
2. For a prospective longitudinal study, why authors do not examine the incidence of depression, anxiety and distress.
3. For a prospective longitudinal study, why authors do not examine the RRs?
4. What is the “Parallel analysis”? I did not found any results from this analysis.
Results
1. Authors should add some subtitles in order to well understanding for readers.
2. Authors should re-design the table 1. A horizontal table is easy to understand. Please turn the first column into the first row.
3. All supplementary tables did not present the titles. Please add them.
4. In all tables. Please correct “p=0.000” to be “p<0.001”.
Discussion
1. As mentioned by authors in the limitation- “Majority of the study population (69.1%) were in the preclinical and paraclinical years which could have influenced the interpretation of results.” Why authors did not control this variable in the association analysis.

---

## Round 0.2 · Major Revisions

Thank you very much for your revisions. Although our reviewers have suggested to accept the paper, I feel that the paper still has much space for improvement.

My first concern is the English language is not written in native American or British English. Further, the paper also has some language issues, for example, line 44 and 52, "&" should not be used, "and" is appropriate. Line 116-117, it is inappropriate to say "depression is found to be higher", which should be the level of depression. I strongly suggest the authors to have their paper polished by native speakers.

My second concern is the unnecessarily long abstract. I think it can be written in a more concise way. For example, there is no need to provide so many statistical details here. In addition, it is not necessary to abbreviate depression as "D", as well as anxiety and stress.

My third concern is the inaccurate conclusion. Because one of the three mental health indicators, depression, did not change significantly after the outbreak, it seems not strict to say "has a negative impact on mental health". The term "mental health" is broad, which should be specific to anxiety and stress based on findings of the present study.
Third, line 86-89, please update the figures accordingly. India has been the second in the world.

Fourth, line 167-170, sample size estimation is described in a confused way. First, it should not be limited to cross-sectional design, cohort design should also be considered. Second, the sample is recruited from an institution, not an infinite population. Please consult a bio-statistician to address this issue.

·

Basic reporting

Thank you for making the suggested revisions.

Experimental design

I was satisfied with this section in the earlier draft.

Validity of the findings

Thank you for making the suggested revisions.

Additional comments

I am satisfied with the author's revisions. Thank you.

Reviewer 2 ·

Basic reporting

The manuscript was written clearly. Sufficient backgrounds were provided. Results and Tables have been modified as recommended.

Experimental design

Study design as been extended and detailed as recommended.

Validity of the findings

The longitudinal study design strengthened the interpretation of the results.

Reviewer 3 ·

Basic reporting

None

Experimental design

None

Validity of the findings

None

Additional comments

I am very satisfied with your revision.

---

## Round 0.3 · accepted · Accept

I am pleased to accept the paper. Thanks for the revisions.

---

## Author Rebuttal · Round 0.3

**Ministry of higher Education**
**Majmaah University**

18.09.2020

Dr Saikarthik Jayakumar
Assistant Professor,
Department of Basic Medical Sciences,
Majmaah University,
Al Zulfi,
Kingdom of Saudi Arabia 15962

Dear Editor,

We thank you for your generous comments on the manuscript and have edited the manuscript to address all concerns.

We believe that the manuscript is now suitable for publication in PeerJ

Thanking You,

Dr Saikarthik Jayakumar
Assistant Professor of Anatomy

On behalf of all authors.

## Editor's comments

*My first concern is the English language is not written in native American or British English. Further, the paper also has some language issues, for example, line 44 and 52, "&" should not be used, "and" is appropriate. Line 116-117, it is inappropriate to say "depression is found to be higher", which should be the level of depression. I strongly suggest the authors to have their paper polished by native speakers.*

As suggested, the manuscript was edited for language correction by native English speakers.

*My second concern is the unnecessarily long abstract. I think it can be written in a more concise way. For example, there is no need to provide so many statistical details here. In addition, it is not necessary to abbreviate depression as "D", as well as anxiety and stress.*

Corrections made as suggested.

*My third concern is the inaccurate conclusion. Because one of the three mental health indicators, depression, did not change significantly after the outbreak, it seems not strict to say "has a negative impact on mental health". The term "mental health" is broad, which should be specific to anxiety and stress based on findings of the present study.*

*Third, line 86-89, please update the figures accordingly. India has been the second in the world.*

Corrections made as suggested.

*Fourth, line 167-170, sample size estimation is described in a confused way. First, it should not be limited to cross-sectional design, cohort design should also be considered. Second, the sample is recruited from an institution, not an infinite population. Please consult a bio-statistician to address this issue.*

The methodology section has been corrected as per suggestions from a biostatistician.